# *LG5*, a Novel Allele of *EUI1*, Regulates Grain Size and Flag Leaf Angle in Rice

**DOI:** 10.3390/plants12030675

**Published:** 2023-02-03

**Authors:** Zhen Li, Junrong Liu, Xingyu Wang, Jing Wang, Junhua Ye, Siliang Xu, Yuanyuan Zhang, Dongxiu Hu, Mengchen Zhang, Qun Xu, Shan Wang, Yaolong Yang, Xinghua Wei, Yue Feng, Shu Wang

**Affiliations:** 1College of Agronomy, Shenyang Agricultural University, Shenyang 110866, China; 2Chinese National Center for Rice Improvement and State Key Laboratory of Rice Biology, China National Rice Research Institute, Hangzhou 311400, China; 3College of Agronomy, Heilongjiang Bayi Agricultural University, Daqing 163319, China; 4National Nanfan Research Institute (Sanya), Chinese Academy of Agricultural Sciences, Sanya 572024, China

**Keywords:** *LG5*, grain size, grain weight, flag leaf angle, rice

## Abstract

Grain size and flag leaf angle are two important traits that determining grain yield in rice. However, the mechanisms regulating these two traits remain largely unknown. In this study, a rice *long grain 5* (*lg5*) mutant with a large flag leaf angle was identified, and map-based cloning revealed that a single base substitution followed by a 2 bp insertion in the *LOC_Os05g40384* gene resulted in larger grains, a larger flag leaf angle, and higher plant height than the wild type. Sequence analysis revealed that *lg5* is a novel allele of *elongated uppermost internode-1* (*EUI1*), which encodes a cytochrome P450 protein. Functional complementation and overexpression tests showed that *LG5* can rescue the bigger grain size and larger flag leaf angle in the Xiushui11 (XS) background. Knockdown of the *LG5* transcription level by RNA interference resulted in elevated grain size and flag leaf angle in the Nipponbare (NIP) background. Morphological and cellular analyses suggested that *LG5* regulated grain size and flag leaf angle by promoting cell expansion and cell proliferation. Our results provided new insight into the functions of *EUI1* in rice, especially in regulating grain size and flag leaf angle, indicating a potential target for the improvement of rice breeding.

## 1. Introduction

Grain size is a major target of rice breeding that is not only a component of yield but also a quality trait. Rice grain yield mainly depends on four major components: grain weight, number of panicles (or tillers) per plant, number of grains per panicle, and the ratio of filled grains [1]. The grain size of rice directly determines the grain weight, which, in turn, affects the yield [2]. Rice grain size consists of grain length, grain width, grain thickness, and grain length-to-width ratio, which is a complex trait controlled by multiple quantitative trait loci (QTLs)/genes. More than 500 QTLs related to rice grain size have been identified on 12 chromosomes. To date, several QTLs/genes controlling grain size have been cloned in rice, such as *GS3*, *GW2*, *qSW5/GW5*, *GS5*, *GL3.1/qGL3*, *GW8*, *TGW6*, *BG2*, *GW6a*, *GL7/GW7*, *GLW7*, *GS9*, *SMG1/OsMKK4*, and *SMG2/ OsMKKK10*; these genes play a very important role in the regulation of grain size in rice [3]. Recent advances have identified several signaling pathways involved in determining grain size, including transcriptional regulatory factors, the ubiquitin–proteasome pathway, G-protein signaling, mitogen-activated protein kinase (*MAPK*) signaling, and phytohormones [3]. For instance, *GS2* encodes the transcription factor *OsGRF4*, which controls grain size by promoting cell expansion and cell proliferation. *GW2* and *OsOTUB15* are two major genes controlling grain width, both of which regulate grain size through the ubiquitin pathway; *GW2* encodes a ubiquitin ligase, which is a negative regulator of grain width, whereas *OsOTUB15* encodes a constitutive ubiquitin-specific protease, which is a positive regulator of grain width [4,5]. The heterotrimeric G-protein complex participates in plant development and consists of *G_α_*, *G_β_*, and *G_γ_* subunits. *GS3* encodes a noncanonical *G_γ_* homologous to *AGG3* [6] and acts antagonistically with another noncanonical *G_γ_*, *GGC2*, to regulate grain size by competitively binding *G_β_* [7]. The *MAPK* cascades are composed of three tiers of protein kinases: an *MAPK* kinase (*MKKK*), an *MAPK* kinase (*MKK*), and an *MAPK* [8]. In rice, *OsMKKK10*, *OsMKK4*, and *OsMAPK6* act as a cascade to regulate grain size [9]. In rice, the grain size genes *GS5* and *qGL3*/*GL3.1* were reported to be involved in the brassinosteroid (BR) signaling pathway [10].

Flag leaf angle is defined as the inclination between the flag leaf blade and the vertical culm of a plant [11]. The flag leaf is the topmost leaf closest to the panicle and plays an important role in determining plant density, photosynthetic rate, and yield potential [12]. The flag leaf angle is primarily determined by the lamina joint structure, which maintains the flag leaf angle. The elongation of cells in the lamina joint, the development of mechanical tissues, and the composition of the cell wall all impact the flag leaf angle [13,14]. Previous studies have shown that the leaf angle is a complex trait controlled by multiple genes involved in several signals in rice, including *ILA1*, *OsBRI1*, *OsSPY*, *OsLG1*, *OsDWARF4*, *D2/CYP90D2*, *OsILI1*, *OsBU1*, *OsVIL3/LC2*, *OsGH3-1/LC1*, *OsBRI1/D61*, *RAV6*, and *OsARF19* [8,13,14,15,16,17,18,19,20,21,22]. The reported genes associated with flag leaf angle have different molecular mechanisms, but there is a common opinion that phytohormone synergism plays a key role in regulating this feature [23]. Brassinosteroid (BR) signaling is a major regulatory pathway that controls flag leaf angle and promotes cell elongation and division the adaxial side of the lamina joint [24,25]. For example, *osdwarf4-1* was reported to function redundantly in C-22 hydroxylation, and the rate-limiting step of BR biosynthesis showed an erect leaf phenotype with enhanced grain yields in rice [16]. Another example, *OsBZR1*, was found to serve as a BR signaling factor, regulating plant height and leaf angle in rice [26]. Gibberellin (GA) has also been reported to be involved in controlling the laminal joint inclination. For instance, *OsSPY* is a negative regulator of GA signaling. Reduced expression of *OsSPY* was reported to lead to elevated laminal joint inclination [17].

Grain size and flag leaf angle are two important agronomic traits that determine grain yield in rice [27]. It was reported that phytohormones and transcriptional factors are associated with these two traits. For instance, *SLG* controls grain size and flag leaf angle by modulating BR homeostasis in rice [27]. *OsARF6* controls flag leaf angle and grain size by responding to auxin [12]. In addition, *OsBUL1* encodes a basic helix–loop–helix (bHLH) transcriptional factor that regulates grain size and flag leaf angle [28]. Nevertheless, the molecular and cellular mechanisms underlying grain size and flag leaf angle remain largely unknown. In the present study, we characterized a mutant *long grain 5* (*lg5*), a novel allele of *elongated uppermost internode-1* (*EUI1*)*. EUI* is well known for regulating internode elongation by modulating GA responses in rice [29]. Interestingly, the novelty of this work is that *LG5* was found to negatively regulate grain size, thousand-grain weight, and flag leaf angle. Our results provide new insight into the modification of seed size and plant architecture.

## 2. Results

### 2.1. The lg5 Mutant Increase Grain Size and Elevate Flag Leaf Angle

To understand the mechanisms governing rice grain size and flag leaf angle, we previously identified a mutant with long grains and a large flag leaf angle in a mutant bank (Figure 1A,F). The *lg5* mutant was isolated from the cobalt 60 radiation-induced M_2_ populations of the Xiushui (XS) mutant bank. The *lg5* mutant showed larger grains compared to XS (Figure 1A,C). The length and width of *lg5* grains were significantly increased compared with those of XS grains (Figure 1B,D). The average length of XS and *lg5* grains was 7.51 ± 0.10 mm and 8.23 ± 0.06 mm, respectively. The average width of XS and *lg5* grains was 3.43 ± 0.06 mm and 3.80 ± 0.10 mm, respectively. In addition, the thousand-grain weight of *lg5* was increased compared with that of XS (Figure 1E). The average thousand-grain weight of XS was 26.97 ± 0.44 g, whereas that of the *lg5* mutant was 29.63 ± 0.26 g. The *lg5* mutant also showed a larger flag leaf angle and higher plant height than XS (Figure 1F,H). The average flag leaf angle of XS and *lg5* was 22.67 ± 2.52° and 60.33 ± 2.52°, respectively (Figure 1G). The average plant height of XS and *lg5* was 85.33 cm and 118.33 cm, respectively (Figure 1I). Furthermore, the lengths of the panicle and internode of every part of *lg5* was increased significantly relative those of XS. The length of the panicle, first internode, second internode, third internode, fourth internode, and fifth internode of *lg5* were increased by 5.30%, 62.78%, 60.93%, 59.76%, 73.27%, and 244.45%, respectively, relative to those of XS (Appendix A). Together, these results indicated that *LG5* influenced grain size and plant architecture in rice.

### 2.2. Map-Based Cloning of LG5

We isolated the *LG5* gene via map-based cloning using an F_2_ population derived from a cross between the *lg5* mutant line and the *indica* rice variety Huasizhan. The F_1_ progeny showed the same phenotype as the XS plants, and 642 of 2554 individual plants in the F_2_ population showed the *lg5* phenotype, in line with a mendelian 3:1 (wild type: mutant) ratio, indicating that the mutant phenotype was controlled by a single recessive gene. We used 171 simple sequence repeats (SSRs) covering 12 chromosomes to locate the *LG5* gene, which was initially mapped to an 847 kb interval of chromosome 5 (Chr. 5) between the SSR markers RM18847 and RM18903 (Figure 2A) and further fine-mapped to a 150 kb interval between the makers RM18883 and RM18893. There are 33 predicted open reading frames. We failed to develop more polymorphic markers to further limit the candidate region, so we sequenced 33 open reading frames and associated them with the known genes between makers RM18883 and RM18893. Comparing the genomic sequences of the candidate region revealed one substitution of A to GT in the *LOC_Os05g40384* gene (Figure 2B) and resulted in premature transcription termination after a 345 bp translation (Figure 2C). These results suggested that the *LOC_Os05g40384* gene was responsible for the *lg5* mutant.

### 2.3. Confirmation of LG5 Function

To confirm whether the *LOC_Os05g40384* gene was responsible for the *lg5* mutant phenotype, we performed a genetic complementation (Com) test by introducing 12,149 bp of *LG5* genomic DNA containing the promoter and the full genomic coding region of *LG5* into *lg5* mutant plants and generated 32 transgenic plants (*LG5*-Com). T_1_ complementary line Com-1 showed a shorter grain length (7.80 ± 0.07 mm), reduced thousand-grain weight (27.46 ± 0.13 g), reduced flag leaf angle (21.00 ± 2.97°), and depressed plant height (80.80 ± 1.23 cm) compared with the grain length (8.22 ± 0.09 mm), thousand-grain weight (29.14 ± 0.24 g), flag leaf angle (70.00 ± 11.18°), and plant height (98.00 ± 2.87 cm) of *lg5*. T1 complementary line Com-2 showed a shorter grain length (7.40 ± 0.09 mm), reduced thousand-grain weight (27.22 ± 0.14 g), lessened flag leaf angle (21.00 ± 3.13°), and depressed plant height (79.10 ± 1.67 cm) compared with *lg5* (Figure 3A,B,D–G), whereas the wild type showed a grain length of 7.52 ± 0.10 mm, a thousand-grain weight of 27.01 ± 0.26 g, a flag leaf angle of 21.00 ± 2.52°, and a plant height of 80.40 ± 3.51 cm. Furthermore, we overexpressed the *LG5* gene driven by the cauliflower mosaic virus 35S promoter in the XS background and generated 40 transgenic plants (*LG5*-OE). We found that 100% of T_0_ transgenic plants were extremely dwarfed and unable to bear grains (Figure 3C). *LG5*-OE plants exhibited a decrease in plant height (Figure 3H). We also examined the *LG5* relative mRNA levels in the culm of XS and *LG5*-OE plants (Figure 3I), and the expression levels of *LG5* were found to be positively correlated with the plant height. To confirm that the phenotype resulted from the *LG5* gene, the RNA interference plants of *LG5* (*LG5*-RNAi) were generated in the Nipponbare (NIP) background. Transgenic plants exhibited increased phenotypes of grain length, grain width, and thousand-grain weight relative to NIP in the mature stage (Figure 4A–E). In addition, transgenic plants produced higher plant height and larger flag leaf angle than NIP (Figure 4F–H). The results reported above are consistent with previous research indicating that dwarfism is caused by elevated expression of *EUI1* [30].

Our results indicated that the *lg5* mutant phenotype was caused by the mutant of the *LOC_Os05g40384* gene, which functioned as a negative regulator of grain size, grain weight, flag leaf angle, and plant height. These findings showed that the *LOC_Os05g40384* was a valuable gene for rice genetics research and breeding, providing new potential for the improvement of seed size and plant architecture.

### 2.4. LG5 and Its Homologs Are Conserved in Cereal Crops

To explore the evolutionary relationship between *LG5* genes and their homologs among six cereal crop species, we conducted an online search (https://phytozome-next.jgi.doe.gov/blast-search) accessed on 6 December 2021 among *Oryza sativa*, *Zea mays*, *Triticum aestivum*, *Sorghum bicolor*, *Hordeum vulgare*, and *Glycine max*. Then, a total of 33 proteins from *Oryza sativa* (1), *Zea mays* (4), *Triticum aestivum* (12), *Sorghum bicolor* (5), *Hordeum vulgare* (5), and *Glycine max* (6) were selected because they had more than 75% identical proteins in common with LG5 protein, and a neighbor-joining phylogenetic tree was constructed using Mega 6. According to the homologous degree of the *LOC_Os05g40384* protein, the 33 proteins could be divided into three major groups—modern, intermediate, and ancient—according to the homologous degree relative to the LG5 protein from highest to lowest (Appendix A). *LG5* was conserved in cereal crops.

### 2.5. LG5 Regulates the Grain Size and Flag Leaf Angle by Controlling Cell Expansion and Proliferation

In rice, grain size is restricted by the size of the spikelet hull [31], which is determined by both cell expansion and proliferation. Therefore, we examined cell size and numbers in the cross sections of the central parts of the spikelet hulls just before heading between the *lg5* mutant and XS (Figure 5A), larger inner parenchyma cells were found in the *lg5* mutant than in XS (Figure 5B,C). The inner parenchyma cells of spikelet hulls in the *lg5* mutant were longer (by 4.80%) and contained more cells (by 6.32%) than those in XS, with an increase in total cell length (11.49%) relative to XS (Figure 5D–F). These results indicated that the expansive grain size of *lg5* contributed to increases in both cell number and cell size. Scanning electron microscopy of the inner and outer surfaces of glumes also showed that *lg5* had a larger cell size than XS (Figure 6A,D) and that the inner spikelet hull surfaces in the *lg5* mutant were longer (by 30.29%) and wider (by 16.11%) than those of XS (Figure 6B,C), whereas the outer spikelet hull surfaces in *lg5* mutant were longer (by 104.56%) and wider (by 9.02%) than those in XS (Figure 6E,F). These results suggested that *LG5* controlled grain size by regulating cell expansion and proliferation in rice glumes.

The leaf lamina joint, which connects the leaf blade and sheath, is the most important tissue governing the leaf angle. It has been found that most of the identified rice mutants with altered leaf inclination arecaused by abnormal expansion and division of changed cells in the leaf lamina joint [20,22,32]. Therefore, we examined leaf lamina joint cells through paraffin sections between XS and *lg5* mutants (Figure 5G). The *lg5* mutant displayed larger cell expansion than XS in the lamina joint (Figure 5H). Paraffin sections showed compact and regular cells in *lg5* compared with XS, resulting in longer (186.58%) and wider (100.85%) cells in *lg5* than XS (Figure 5I,J).

Several genes were found to control grain size by influencing cell proliferation and cell expansion processes, such as *GS2*; *GL7* was found to be involved in the regulation of cell expansion; and *GW2*, *GL3.1*, *TGW6*, and *GS5* were found to control grain size by regulating cell proliferation. To reveal how *LG5* regulates cell expansion and cell proliferation in spikelet hulls, we detected its expression levels in the XS and *lg5* young panicles. Compared with XS, *GS2* and *GS5* expression was significantly increased in the *lg5* mutant (Figure 6G), suggesting that the increase in cell size and number of *lg5* might result from the elevated expression of *GS2* and *GS5* because they are positive regulators of grain size [33,34]. The expression levels of *TGW6*, *GL7*, *GW2*, and *GL3.1* were similar between XS and *lg5* mutants according to reverse transcription polymerase chain reaction (RT-PCR) results.

We also detected the expression of genes involved in the cell cycle, such as *CYCA1;1*, *CYCA3;2*, *CYCB1;1*, *CYCB2;2*, *CDKA2*, and *CDKB2;1*. Compared with XS, the six cell-cycle-related genes were all upregulated in the *lg5* mutant (Figure 6H). These results indicated that *LG5* may regulate grain size by promoting cell proliferation through enhanced expression of several cell-cycle-related genes.

### 2.6. The Expression Pattern and Subcellular Localization of LG5

To examine the temporal and spatial expression pattern of *LG5*, total RNA from eight organs from XS plants, including root, tiller bud, stem, node, pulvinus, leaf sheath, flag leaf, and young panicle, were extracted. We examined the expression of *LG5* using quantitative real-time RT-PCR analysis. The results revealed that *LG5* was constitutively expressed in various rice organs and especially highly expressed in the flag leaf, young panicle, and leaf sheath (Figure 7A). A similar study also reported that the highest *LG5* expression occurred in young panicles and internodes [29]. To further determine the involvement of *LG5* in different stages of young panicles, we examined the expression of *LG5* in four stages of spikelet (young 2 cm, 7 cm, 13 cm, and 20 cm panicles). We found that *LG5* was most highly expressed in the young 2 cm panicle (Figure 7B). This result was consistent with a previous study of *EUI1*-GUS, which was detected mainly in rapidly elongating or dividing tissues [29]. Those results showed that *LG5* played an important role in regulating grain size, especially in the young panicles.

To examine the subcellular localization of *LG5*, we constructed *LG5*::GFP and *HDEL*-mCherry fusions driven by the 35S promoter. The two fusion protein constructs were transferred together into tobacco and rice protoplasts. We observed that the *LG5*::GFP and *HDEL*-mCherry fusion proteins were localized in the endoplasmic reticulum both in tobacco and rice protoplasts (Figure 7C,D). These results indicated that *LG5* encoded an endoplasmic reticulum protein, consistent with previous evidence [29].

## 3. Discussions

Grain size and flag leaf angle are two important agronomic traits that influence grain yield in rice. Many genes associated with these two traits have been identified and characterized, but the molecular and genetic mechanisms underlying grain size and flag leaf angle remain largely unknown [3,11,35]. In the present study, a mutant with a large grain and flag leaf angle, named *lg5*, was identified and characterized (Figure 1). Comparison of the genomic sequences of the candidate region indicated that *LOC_Os05g40384*/*EUI1* is the most likely candidate gene for *lg5* (Figure 2). A complementation test restored the grain size and flag leaf angle to the wild-type phenotype. Overexpression of *LG5* sharply reduced plant height to a dwarf size and eliminated the ability to bear grains in the XS background (Figure 3). These results confirmed our assumption that the *LOC_Os05g40384* gene was the target gene. In our study, *LG5* was found to be involved in regulating important agronomic traits including grain size, flag leaf angle, and plant height. To investigate the function and application of the *LG5* gene, we generated *LG5*-RNAi lines in the NIP background. Some agronomic traits such as grain size, flag leaf angle, and plant height were enhanced in the *LG5*-RNAi lines (Figure 4). Our results indicated the possibility of applying the *LG5* gene to modify seed size and plant architecture. *EUI1* (elongated uppermost internode) produced a near doubling in the length of the uppermost internode [36]. To date, some evidence has been found that the *EUI1* gene could affect multiple traits in rice; it was found to regulate disease resistance to bacterial blight and blast resistance in rice [37] and to play a role in root gravity responses in rice [38]. Interestingly, in our study, we found that *EUI1* also acted as a negative regulator of grain size and flag leaf angle, providing new insights into the functions of *EUI* and supplying valuable germplasm resources for rice breeding.

Cell expansion and cell proliferation determined organ size [39,40,41]. In our study, cell length and total cell numbers were significantly increased in *lg5* compared with those in XS in the cross sections of spikelet hulls (Figure 5A–F), and the length and width of epidermal cells of the inner and outer glumes were also increased in *lg5* compared with those in XS (Figure 6A–F). To determine the possible regulatory relationship between *LG5* and other identified genes that control grain size by regulating cell expansion, such as *GS2*, and *GL7*, we examined the transcript level of these genes and found that the expression level of the *GS2* gene was significantly increased in the *lg5* mutant. In addition, *TGW6*, *GW2*, *GL3.1*, and *GS5* were found to control grain size by regulating cell proliferation, and the transcript level of *GS5* was significantly increased in the *lg5* mutant (Figure 6G). These results suggested that *LG5* may control grain size by regulating the expression of the cell expansion gene and cell proliferation gene. Furthermore, cell cycle genes played functional roles in the cell cycle regulation of seed size and development in plants [42,43]. In this study, several cell cycle genes, including two A-type cyclins (*CYCA1;1* and *CYCA3;2*), two B-type cyclins (*CYCB1;1* and *CYCB2;2*), and two cyclin-dependent kinases (*CDKA2* and *CDKB2;1*), were upregulated in the *lg5* mutant (Figure 6H). The upregulation of these cell-cycle-related genes may contribute to cell division in the spikelet and lead to expanding grain size. These observations suggested that *LG5* may control grain size by regulating the expression of cell cycle genes. Histological analysis of the leaf lamina joints, which exhibited significant increases in cell size in *lg5* relative to those in XS (Figure 5G–I), indicated that *LG5* may control flag leaf angle by regulating cell expansion in leaf lamina joints. This finding was consistent with literature published during past decade study, indicating the *EUI1* gene has a positive effect on parenchyma cell division and elongation [44].

Grain size and leaf angle are critical agronomic traits that determine final yields [27], and both are regulated by phytohormones [3,23]. In rice, BR played important role in the regulation of grain size, leaf angle, plant height, and tiller number [45,46]. For example, overexpression of *OsBIM1*, which was involved in BR signaling, could significantly increase rice leaf angles but decrease grain size and reduced yield [47]; knockout of *OsBUL1*, which was involved in BR signaling through regulation of HLH proteins, produced erect leaves with small grains [48]. GA was another predominant hormone regulating plant cell elongation and always exhibits crosstalk with BR in complex mechanisms [49]. GA signaling-related genes simultaneously controlled grain size and leaf angle and were evident in the case of *GW6/OsGSR1*, which was a positive regulator of GA signaling. *GW6/OsGSR1* could activate BR synthesis through direct interaction with the BR biosynthesis enzyme, *DWF1*, and the *GW6/OsGSR1*-RNAi lines showed erect leaves and decreasing seed size [50]. In our study, an *lg5* mutant frameshift site occurred in the P450 binding domain region, which resulted in the loss-of-function mutant allele of *Os05g40384*. Additionally, *EUI1* encoded a cytochrome P450 monooxygenase CYP714D1, which was associated with modulating GA responses [29]. Therefore, *LG5* might be involved in GAs to coregulate grain size and flag leaf angle. A previous study found that *EUI1* was associated with the loss of *SLR1* during GA signaling and played a negative regulatory role in GA-mediated cell elongation [51,52]. *EUI1* was involved in GA homeostasis not only in the internodes during the heading stage but also in the roots and seeds [38]. Our findings illustrated that *LG5* might regulate grain size and flag leaf angle by participating in the phytohormone pathway. In summary, we characterized a mutant *lg5*, which probably functioned in the phytohormone pathway and contributed to increased grain size, thousand-grain weight, flag leaf angle, and plant height in rice.

## 4. Materials and Methods

### 4.1. Plant Materials and Growth Conditions

The *japonica* rice XS was radiated with 60 Co-γ. We selected the mutant *lg5* with a long grain and flag leaf angle in the M_2_ generation and further confirmed the phenotype in M_3_ and M_4_ progenies. Stable inheritance M_3_ and M_4_ were used in this research. The *lg5* mutant and XS were planted in Hangzhou and Hainan in four generations over two years, and the characters of *lg5* were stable in different environments. The *lg5* mutant was crossed with the *indica* rice Huasizhan to generate an F_2_ segregation population for gene mapping. All the parents, F_1_ plants, and F_2_ individuals used for morphological and genetic analyses were grown in paddy fields under natural conditions at the China National Rice Research Institute (Hangzhou, Zhejiang Province).

### 4.2. Map-Based Clone of LG5

The mutant individuals from the F_2_ generation of a cross between the *lg5* mutant and Huasizhan were selected for mapping. Bulked segregant analysis was used to rapidly locate the mutation as follows: equal amounts of flag leaf from each of 20 XS plants and 20 *lg5* mutant plants were sampled for DNA extraction to form an XS pool and an *lg5* pool. The parents and the two DNA pools were subjected to preliminary linkage analysis by genotyping 171 polymorphic SSR markers covering 12 chromosomes. Subsequently, 642 *lg5* mutant individual plants from the F_2_ population were genotyped to determine the physical location of *LG5*. Sequences of primers for mapping are listed in Appendix A.

### 4.3. Phenotyping

The seeds were individually harvested for phenotypic investigation during the mature stage. Fully filled grains were used to measure grain length, grain width, and thousand-grain weight. The grain length, grain width, and thousand-grain weight were evaluated using an automatic seed counting and analyzing instrument (Model SC-G, Wanshen Ltd., Hangzhou, China). Each measurement was repeated three times, and the mean value was recorded. Flag leaf angles were measured by a protractor. Plant heights were measured by a ruler during the mature stage.

### 4.4. Vector Construction and Plant Transformation

We sequenced the entire genomic DNA region of *LG5* in XS and NIP. XS and NIP had the same sequence of the *LG5* gene. To construct a complementary vector, the *LG5* gene containing the 1758 bp promoter region, entire genomic DNA region (9904-bp), and 783 bp downstream of *LG5* was cloned into the vector PCAMBIA1300 (PC1300) by seamless cloning. To generate the overexpression lines, the 1734 bp CDS of *LG5* was cloned into the vector PCAMBIA1300-CaMV35S (PC1300S), and PC1300 and PC1300s were digested with Kpn1 and Xba1 by double-enzyme digestion. To generate the RNAi lines, the vector pTCK303 was used as described by Wang et al. [53]. To avoid disturbing other homologous genes, 500 bp CDS of *LG5* was used as the *LG5*-RNAi fragment, which was cloned to pTCK303 by BamH1 and Kpn1 for the sense strand and Spe1 and Sac1 for the antisense strand. Genetic transformations were conducted using rice embryogenic calli through agrobacterium tumefacien-mediated transformation [54]. The primers mentioned above are listed in Appendix A.

### 4.5. Subcellular Localization of LG5

To determine the subcellular localization of *LG5*, the 1734 bp CDS of *LG5* was cloned into the vector pYBA1132 containing the GFP gene. The Pro35S::*LG5*-GFP plasmid was introduced into GV3101 (Weidi, AC1003) and injected into young tobacco, whereas the Pro35S::*LG5*-GFP plasmid was introduced into rice protoplasts. The endoplasmic reticulum protein *HDEL* (*LOC_Os05g45310*) was cloned into the vector pYBA1138 fused with the mCherry gene, which was used as an endoplasmic reticulum marker. GFP and mCherry fluorescence signals were observed under a Zeiss LSM710 confocal laser-scanning microscope (Carl Zeiss AG, Jena, Germany). The primers mentioned above are listed in Appendix A.

### 4.6. Phylogenetic Tree Analysis

To create a meaningful phylogenetic tree comparing homogenous *LG5* in *Oryza sativa*, *Zea mays*, *Triticum aestivum*, *Sorghum bicolor*, *Hordeum vulgare*, and *Glycine max*, the proteins of the 33 varieties were compared. A FASTA file containing a list of proteins was downloaded from the phytozome database for each protein. Phylogenetic evolutionary tree analysis was performed by Mega 6 using the neighbor-joining method and modified online (https://www.evolgenius.info/evolview) accessed on 6 December 2021.

### 4.7. Paraffin Section Analysis

Cross sections of the rice glumes and flag leaf lamina joint areas were analyzed by paraffin sectioning. They were fixed in FAA (50% ethanol, 5% formaldehyde, and 5% glacial acetic acid) for more than 48 h. The fixed samples were dehydrated in a graded ethanol series (30, 50, 75, 95, 100, 100, and 100%), cleared in a xylene series (50, 70, 90, 100%, and 100%), and embedded in paraffin. Cross sections were stained with toluidine blue, stored in paraffin, cut with the YD-335B microtome (Shanghai Zhisun Equipment Co. Ltd, Shanghai, China), and observed by the S-3000N scanning electron microscopy (Hitachi, Tokyo, Japan). They were operated as previously described by Ruan et al. [55].

### 4.8. Transmission Electron Microscopy Analysis

For rice glume cell observation, the spikelets of the *lg5* mutant and XS were collected as samples during the maturity stage. The samples were fixed in FAA solution (formalin: glacial acetic acid: ethanol in a 1:1:18 ratio by volume) at 4 °C for 24 h, and dehydrated and dried as described by Feng et al. [56]. The inner and outer surfaces of the spikelet glumes were observed under the S-3400 scanning electron microscope (Hitachi, Tokyo, Japan) at Zhejiang University.

### 4.9. RNA Extraction and qRT-PCR Technology

Total RNA was extracted from various rice tissues using a Mini-BEST plant RNA extraction kit (TAKARA, Tokyo, Japan). First-strand cDNAs were synthesized using Prime Script RT Master Mix (TAKARA, Tokyo, Japan). Quantitative RT-PCR analysis was performed on an Applied Biosystems 7500 real-time PCR system (Invitrogen, Carlsbad, CA, USA) with a 2 × SYBR Green PCR Master Mix (Applied Biosystems, Foster, FL, USA). The rice *ACTIN* or *UBIQUITIN* gene was used as an internal control. Each measurement was replicated at least three times with three biological samples. The RT-PCR primers used in these assays are listed in Appendix A.

## Figures and Tables

**Figure 1 plants-12-00675-f001:**
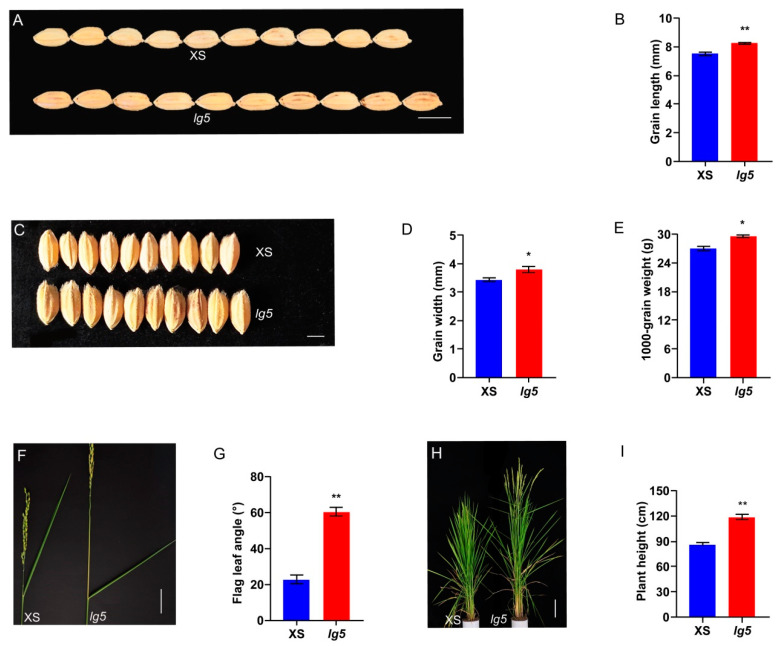
Comparison of agronomic traits between XS and *lg5* in the mature stage. (**A**,**B**) Grain length of XS and *lg5*; bar = 8 mm. (**C**,**D**) Grain width of XS and *lg5*; bar = 3.5 mm. (**E**) Thousand-grain weight of XS and *lg5*. (**F**,**G**) Flag leaf angle of XS and *lg5*; bar = 5 cm. (**H**,**I**) Plant height of XS and *lg5*; bar = 15 cm. Data are presented as means ± SD. * and ** indicate *p* < 0.05 and *p* < 0.01, respectively, according to Student’s *t*-test.

**Figure 2 plants-12-00675-f002:**
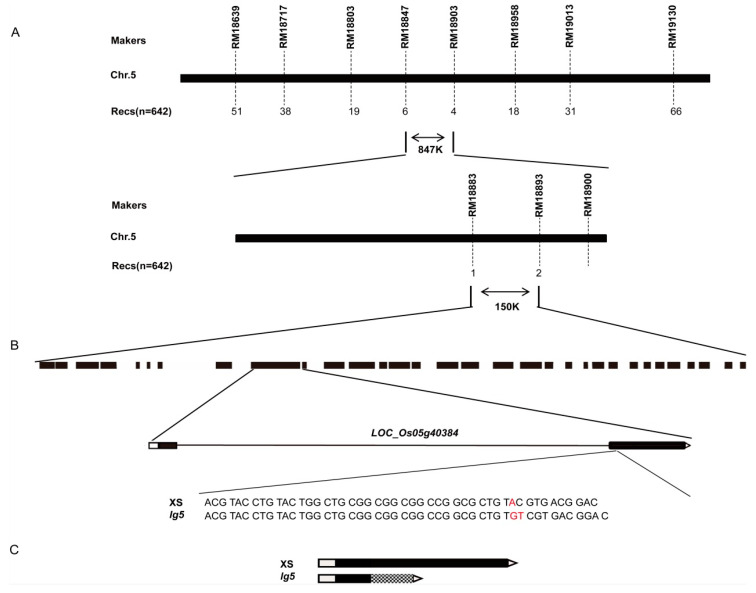
Map-based cloning of *LG5*. (**A**) Initial map and fine mapping of *LG5;* the numbers beneath the marker positions indicate the number of recombinants (recs). (**B**) A total of 33 candidate genes and *LOC_Os05g40384* gene substitution (depicted in red). (**C**) *LOC_Os05g40384* coding sequence (CDS) between XS and *lg5*; the shaded area represents the changed translation and premature termination.

**Figure 3 plants-12-00675-f003:**
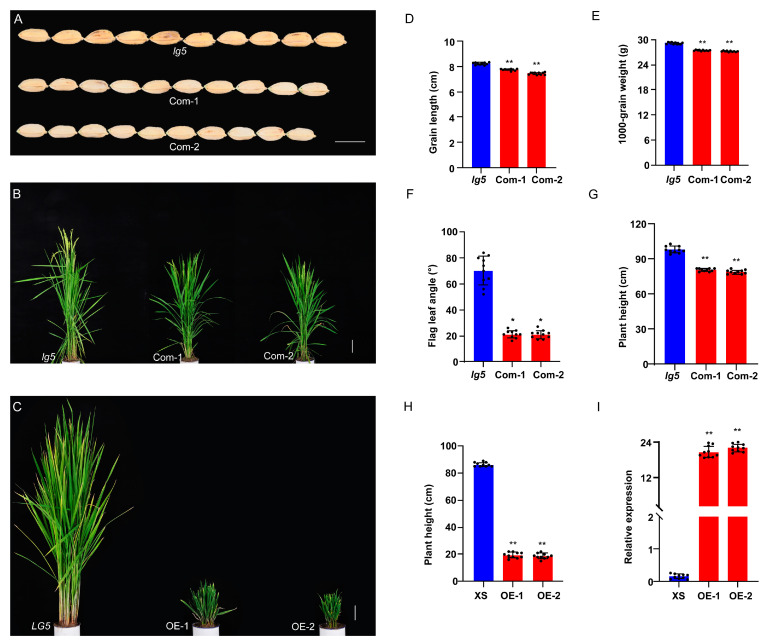
Complementary test and overexpression (OE) analysis at the mature stage. (**A**) Grain length of *lg5* and Com; bar = 8 mm. (**B**) Plant height of *lg5* and Com; bar = 10 cm. (**C**) Plant height of XS and OE; bar = 10 cm. (**D**–**G**) Grain length, thousand-grain weight, flag leaf angle, and plant height of *lg5* and Com. (**H**) Plant height of XS and OE. (**I**) Relative expression of *LG5* in culms of XS and OE with the rice *ACTIN* gene used as an internal control. Data are presented as means ± SD. * and ** indicate *p* < 0.05 and *p* < 0.01, respectively, according to Student’s *t*-test.

**Figure 4 plants-12-00675-f004:**
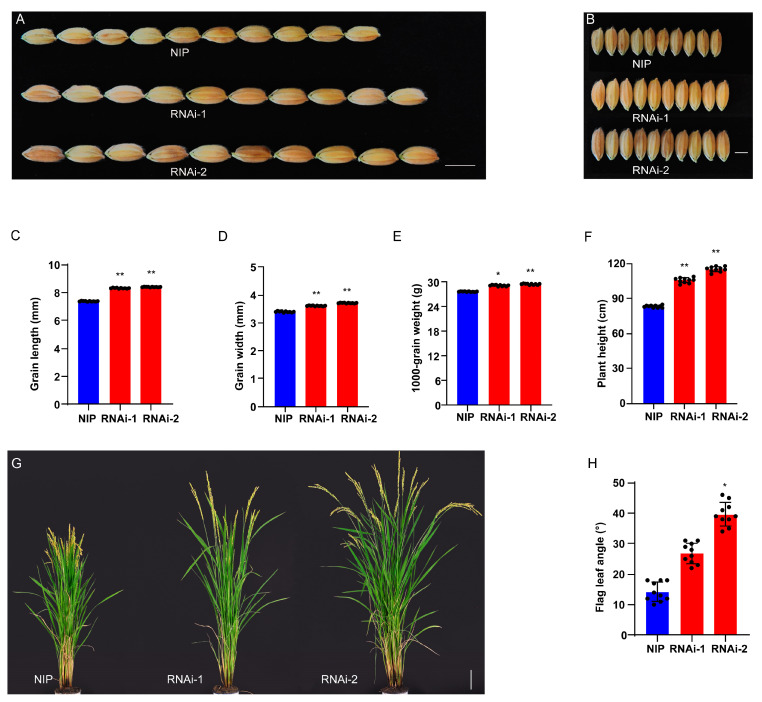
RNAi test at the mature stage. (**A**) Grain length of NIP and *LG5-*RNAi; bar = 6 mm. (**B**) Grain width of NIP and *LG5-*RNAi; bar = 3.5 mm. (**C**–**E**) Grain length, grain width, and thousand-grain weight of NIP and *LG5-*RNAi. (**F**–**H**) Plant height and flag leaf angle of NIP and *LG5-*RNAi; bar = 10 cm. Data are presented as means ± SD. * and ** indicate *p* < 0.05 and *p* < 0.01, respectively, according to Student’s *t*-test.

**Figure 5 plants-12-00675-f005:**
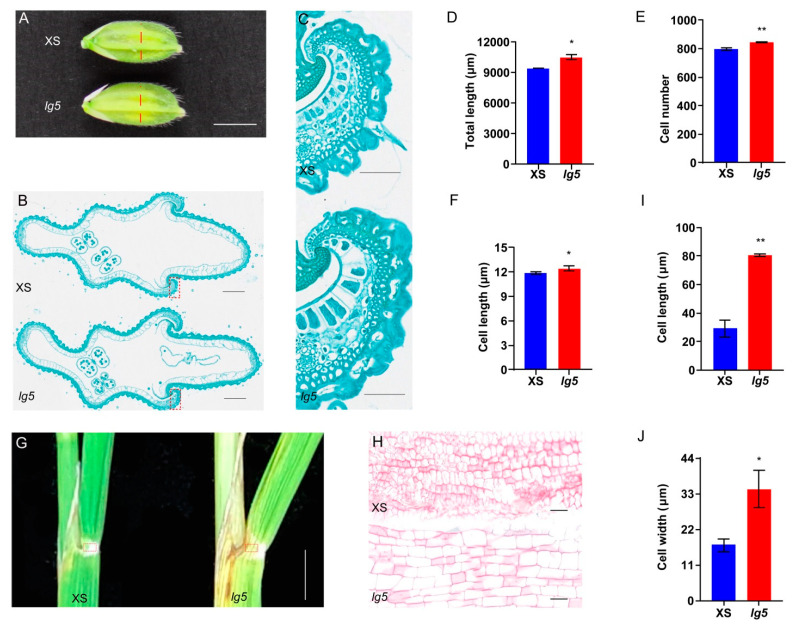
Paraffin sections of XS and *lg5*. (**A**) Spikelet glume just before heading; bar = 3.4 mm. (**B**) Central parts of spikelet hulls of the line in (**A**); bar = 320 μm. (**C**) Magnified views of the boxed areas of (**B**); bar = 60 μm. (**D–F**) Total length, cell number, and cell length in the outer parenchyma layer of the spikelet hulls. (**G**) Comparison of leaf lamina joint; bar = 5 mm. (**H**) Magnified views of the boxed areas of (**G**); bar = 50 μm. (**I**,**J**) Cell length and cell width of the leaf lamina joint. Data were presented as means ± SD. * and ** indicate *p* < 0.05 and *p* < 0.01, respectively, according to Student’s *t*-test.

**Figure 6 plants-12-00675-f006:**
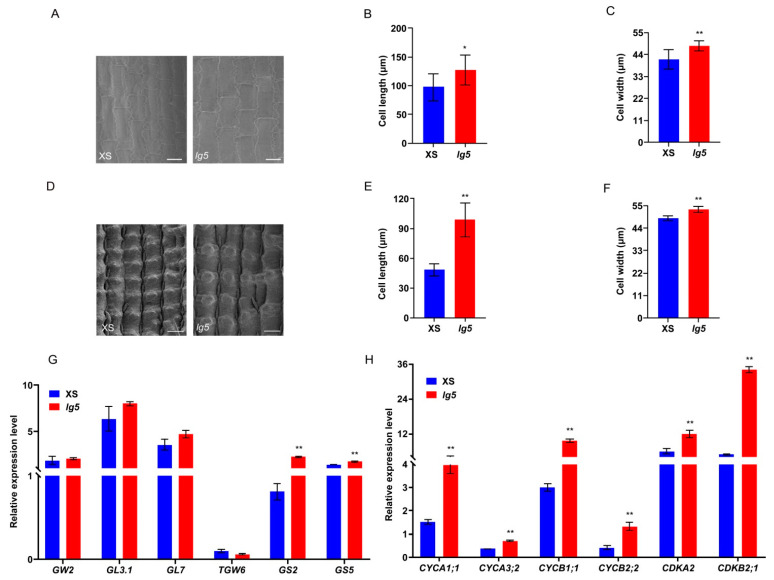
Scanning electron microscope and RT-PCR analysis of XS and *lg5*. (**A**–**C**) Cell length and width of the inner grain hulls; bar = 50 μm. (**D**–**F**) Cell length and width of the outer grain hulls; bar = 50 μm. (**G**,**H**) Expression of cell expansion and cell proliferation genes and expression of cell cycle genes in young panicles, with the rice *UBIQUITIN* gene as an internal control. Data are presented as means ± SD. * and ** indicate *p* < 0.05 and *p* < 0.01, respectively, according to Student’s *t*-test.

**Figure 7 plants-12-00675-f007:**
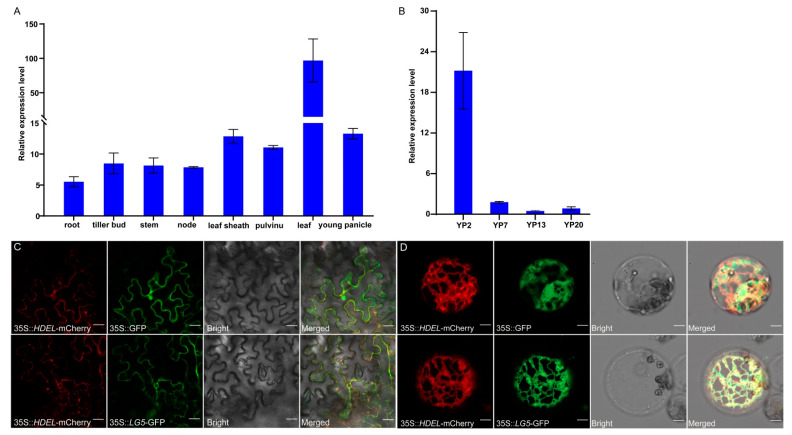
Expression pattern and subcellular localization of LG5 protein. (**A**) Relative expression levels in different tissues in XS. (**B**) Relative expression levels in young 2 cm (YP2), 7 cm (YP7), 13 cm (YP13), and 20 cm (YP20) panicles in XS. (**C**) Subcellular localizations in young tobacco leaf epidermal cells; bar = 20 μm. (**D**) Subcellular localizations in rice protoplasts; bar = 5 μm.

## Data Availability

All data and conclusions are included in this paper.

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
