# Peer review of "LG5, a Novel Allele of EUI1, Regulates Grain Size and Flag Leaf Angle in Rice"

_plants, 2023, doi:10.3390/plants12030675_

Round 1
Reviewer 1 Report
Li et al used map-based marker-assisted cloning method to identify Os05g40384 for the lg5 mutant. This LG5 allele bears a single base change with 2bp insertion, which resulted in a frameshift with a premature stop codon.
First, the abbreviation for gene names needs to be spelled out in full when they are first time mentioned. The EUI1 gene needs to be spelled out.
Second, the authors need to be very careful when mentioning gene names. The authors need to distinguish among using (1) Os05g40384,(2)EUI, and (3) LG5. In the manuscript, the phenotype of the lg5 mutant was the result of the genetic condition of the EUI/Os05g40384 locus. This LG5 locus has a frame-shift mutation which renders the LG5 encodes a foreign protein. This foreign protein only shares a short portion identical to the EUI/Os05h40384 protein. So, when the authors mentioned "over-expressing the LG5", I was wondering if the mutant foreign protein was expressed.
Third, is there a loss-of-function mutant allele for Os05g40384? If not, since the authors have cited several previously published EUI1 related paper, does this lg5 mutant phenotype match what is predicted from those cited paper for its loss-of-function GA-related phenotype? I searched "gibberellin" and found nothing in the discussion.
Fourth, we all know that seed size varies a lot even harvested from the same plant. Just ten a little bigger seeds do not actually mean anything. Figs. 1A, 1C and 3A, etc, should be moved to supplemental material.
Fifth, the authors need to follow the general scientific writing principles. "CV" is a weird expression when referring to a binary vector containing the wild-type genetic copy, I couldn't find any other literature using this. For example, in the figure legend of Fig.3A, "grain length of lg5 and CV"? The figure legend should be self-evident, no one would know what "CV" is without reading the main text.
Sixth, in Figs. 3DEFG, a wild-type XS control is missing. Also, just two independent rescue lines are far from sufficient, besides the authors need to show the expression levels of EUI in these two rescue lines. The authors need to present a scatter box figure, with multiple independent rescue lines, to demonstrate the general trend of the over-expression EUI1.
Seventh, what is wrong with your Fig. 4 in the pdf file?
Last, for Figs. 5 and 6, back to my previous question, are you examining a EUI1 loss-of-function line? Is this lg5 allele a loss-of-function allele? Is such a null allele existing in rice? If yes, do the data in Figs. 5 and 6 share similar results with other null alleles? If not, do these same data match what is predicted for a null allele? From the citations, I could see this seemingly is highly related to GA; however, the authors did not test the mechanism in any of their experiments, and did not discuss this at all.
Reviewer 2 Report
This manuscript reports a molecular genetic study on a cytochrome P450 (EUI1) in rice. 60Co radiation-mutant lg5, which exhibited increased grain size and elevated flag leaf angle, was analyzed to find the dual-function of EUI1 in the LG5 locus for regulating grain size and flag leaf angle. EUI1 seems previously characterized by Zhu et al (2006), so the novelty of this study is with the functions of EUI1. Overall, the main text is clearly written, while it would need to describe the novelty of this study clearly by introducing the study of Zhu et al (2006) in the Introduction section. My concerns about this study were listed as follows:
(1) How the authors found the mutant lg5? The isolation procedures were not given in the manuscript. There are key points to describe scientifically soundness in mutant isolation; (a) which generation of 60Co radiation-mutants were obtained from the XS mutant bank? (b) No any reference or website information on the XS mutant bank? (c) in which generation the phenotyping and screening were performed for the isolation of the original lg5? (d) What about the mutant trait dependency on the environmental effects?
(2) There are unfamiliarity of scientific vocabularies throughout the text. For example, in Line 142 and 531, a word “Map Cloning” is used. The common word that the authors want to express is “Map-based cloning” or “Positional cloning”.
(3) In the Introduction section, many QTL/genes controlling grain size were listed. In the same manner, several genes for controlling leaf angle were listed. I agree with showing the references but wonder if these all QTL/gene names have to be listed in the main text? I feel that just listing names are descriptive.
(4) In Line 297, only “Soybean” is shown as a common name. Why?
(5) In Line 296, how the 33 proteins were selected? In Line 299, how the three major groups and their names were defined? Please describe scientifically in the main text.
(6) In Line 465, there is a sentence which lacks scientific neutrality “We were interested in the application in improving rice breeding”.
(7) There are writing errors. In Line 17 “leaf angle was identified, and map-based cloning…”; in Line 24 “Furthermore” is no need; in Line 34 “of rice” is no need; in Line 109 “59.76%, 73.27%, 244.45%,”; in Line 110 “LG5 influences grain size”; in Line 298 “LG5(non-italic) protein”, in Line 422 “fusion protein constructs were”; in Line 424 “fusion proteins were localized”.
(8) qRT-PCR reference genes were referred by a reference [34]. Is this reference correct? This reference offers the validity of qRT-PCR in rice? There are several papers that provide evidence of gene expression stability of representative reference genes in rice.
(9) The reference list format is all correct? In Line 648, “Theor. Appl Genet”. In Line 766, “Oryza sativa (in italic) and Oryza minuta (in italic)”
(10) Table S1 to S3 lists primer information used in this study. It could be integrated into one Supplementary Table. The element of the table has errors; the 1st column “Primers” is improper, “Forward 5’ – 3’”, “Reverse 5’ – 3’”. Why primer sequences contain letters? Only capital is fine.
Round 2
Reviewer 1 Report
For the revision, the authors have generally successfully answered all my previous concerns. I have no other concerns and I think the revision should be considered for publication.